# Burnout among Health Care Professionals during COVID-19

**DOI:** 10.3390/ijerph191811807

**Published:** 2022-09-19

**Authors:** Siw Tone Innstrand

**Affiliations:** Department of Psychology, Norwegian University of Science and Technology, 7491 Trondheim, Norway; siw.tone.innstrand@ntnu.no

**Keywords:** COVID-19, burnout, health care professionals

## Abstract

The present study examined organizational, situational (i.e., COVID-19-related), and psychological factors associated with burnout during the COVID-19 pandemic among 268 health care professionals in Norway. A total burnout score based on the Burnout Assessment Tool (BAT), the four core BAT subscales (i.e., Exhaustion, Mental Distance, Cognitive Impairment, and Emotional Impairment), and the COVID-19 Burnout Scale served as the dependent variable. Among the results, organizational factors such as work–home conflict, workload, and role conflict were positively related to burnout. Although autonomy and colleague support were negatively related to burnout, support from leaders was positively related to it, which might suggest a suppressive effect. Organizational factors explained most of the variance in general burnout (i.e., BAT Total), whereas situational (i.e., COVID-19-related) factors (e.g., involvement with COVID-19, fear of COVID-19, and COVID-19-induced stress) seemed to better explain COVID-19 burnout. COVID-19-oriented actions were related only to Mental Distance. Psychological factors such as meaning were negatively related to BAT Total, Exhaustion, and Mental Distance, whereas a breach of the psychological contract was related to all subscales. Such results suggest that organizational and situational factors contribute differently to general and COVID-19 burnout and that administering pandemic-specific assessment tools can clarify how the pandemic has affected mental health.

## 1. Introduction

As workers on the front lines of action against COVID-19, health care professionals (HCPs) have faced severe psychological stress and a high risk of mental disorders [1,2]. In fact, as per a recent review, HCPs have shown higher levels of anxiety, stress, depression, and insomnia than other occupational groups during the COVID-19 pandemic [3], primarily due to the risk of infection and concern about infecting family members but also due to the lack of protective equipment, overly high workloads, a lack of rest, and exposure to traumatic life events, including death.

Although a review from before the COVID-19 pandemic also revealed the high, long-standing prevalence of burnout among HCPs [4], that trend accelerated rapidly with the outbreak of COVID-19 [2]. In 2020, it was suggested that approximately one of every three physicians was experiencing burnout at any given time [5]. Burnout is considered to be a major problem for HCPs, one with significant consequences for individual workers as well as patients, their families, organizations, and society at large. For example, a recent review showed that burnout is associated with worsening safety and quality of care, decreased satisfaction among patients, and, in turn, limited organizational commitment and productivity among nurses [6].

Research on the severe acute respiratory syndrome (SARS) that spread in 2003 has shown that both long-term psychological effects (i.e., higher levels of burnout, psychological distress, and post-traumatic stress) and occupational effects (i.e., reduced contact with patients, reduced work hours, maladaptive behavior, and absences due to sickness) continued to impact HCPs working with SARS patients 1–2 years after the outbreak [7]. Thus, the potential long-term consequences of pandemics, including the ongoing COVID-19 pandemic, on HCPs must be taken seriously.

Considering the predicted global shortage of nurses in the next 10–20 years [8] and the probability of more frequent, more severe pandemics in the future [9], research on what causes burnout among HCPs during crises such as the COVID-19 pandemic is essential. Additional knowledge on the triggers of burnout among HCPs during pandemics would be especially useful for efforts to not only provide effective staff support and training in preparation for future outbreaks but also curb the current pandemic’s potential long-term effects.

In a study on burnout among HCPs in 60 countries, Morgantini [2] found higher levels of burnout among HCPs in high-income countries than among their peers in low- and middle-income countries. HCPs in Norway, however, were not included in that study. To clarify the pandemic’s global effect on mental health studies from various countries are needed. Thus, the present study, with a sample of HCPs in Norway, investigated potential COVID-19-related triggers of burnout as well as organizational and psychological factors known to be risk factors for mental disorders among HCPs.

### 1.1. Burnout

According to the World Health Organization’s (WHO) current International Disease Classification (i.e., ICD 11), *burnout* is a “syndrome” that results from “chronic workplace stress that has not been successfully managed” [10]. Despite various assessments and conceptualizations of burnout—for instance, some view it as a one-dimensional construct, others as a multidimensional one [11]—the consensus seems to be that exhaustion is the essential dimension of burnout [12]. In this study, burnout was conceptualized and measured according to two divergent assessments.

On the one hand, the Burnout Assessment Tool (BAT), developed by Schaufeli [12], defines burnout as “a work-related state of exhaustion that occurs among employees, which is characterized by extreme tiredness, reduced ability to regulate cognitive and emotional processes, and mental distancing” (p. 4). In view of the BAT’s novelty, researchers have been called to investigate the usefulness of discriminating between its four subscales dimensions (i.e., Exhaustion, Mental Distance, Cognitive Impairment, and Emotional Impairment), as they could potentially correlate differently with other constructs such as job demands and on-the-job resources [12]. Thus, this study measured both BAT total scores and scores for the BAT’s four subscales.

On the other hand, this study also used a COVID-19-specific measure of burnout adopted from the Burnout Measure-Short Version [13], a one-dimensional scale of burnout based on 10 symptoms of physical, emotional, and mental exhaustion. This choice was based on the suggestion the COVID-19 pandemic has severely impacted the mental health of employees in general and HCPs in particular [11]. Thus, to better understand what factors have affected HCPs’ mental health during the pandemic, using a pandemic-specific assessment seemed necessary [14].

### 1.2. Triggers of Burnout

One of the most-applied theoretical frameworks explaining the path to burnout is the job demand–resource (JD-R) theory, a framework introduced in 2001 [15]. The JD-R model suggests that burnout can arise when demands (e.g., work overload and role conflicts) exceed resources (e.g., autonomy and support from supervisors and/or colleagues) over time [11]. What those demands and resources consist of, however, may vary across different occupations and situations, especially extraordinary ones such as the COVID-19 pandemic.

Maslach and Leiter [16] identified six factors within employee–job mismatches that can lead to burnout: workload, lack of control, insufficient reward, breakdown in community, lack of fairness, and value conflicts. This aligns with Schaufeli et al.’s [12] findings that work–life conflict, interpersonal conflict, and role conflict are the strongest predictors of burnout. Moreover, those mismatches corroborate the findings of a review of 25 years of research addressing burnout among nurses prior to the COVID-19 pandemic [4], particularly that the types of tasks, the organization of those tasks, and relationships with colleagues, supervisors, and/or clients are potential triggers of burnout. More recent qualitative results from China suggest that nurses have experienced increased collective power and team cohesion during the pandemic due to helping each other and providing support in relieving stress [17]. Thus, in this study, it was hypothesized that the following organizational demands and resources are related to burnout:
**Hypothesis** **1.***The organizational demands of (a) work–home conflict, (b) workload, and (c) role conflict are positively correlated with burnout.*
**Hypothesis** **2.***The organizational resources of (a) autonomy, (b) support from colleagues, and (c) support from leaders are negatively correlated with burnout.*

While at the forefront of the fight again the COVID-19 pandemic, most HCPs have become exceptionally aware of the consequences of being infected or other affected by the disease. Although risk perception may be an important factor of protecting oneself from such harm, being too close to or too involved with a risky event threatening COVID-19 infection can create fear and risk mental disorder [18]. Investigating involvement in risky events and risk perception during the COVID-19 outbreak in China, Qian and Li [18] found that involvement with COVID-19 was positively related to mental health in terms of fear and anxiety.

By extension, a recent meta-analysis demonstrated that fear of COVID-19 has indeed been associated with a wide range of mental health conditions, including depression, anxiety, stress, distress, traumatic stress, and insomnia [19]. findings In addition to more fear and anxiety Sun et al. [17] found that caregivers also reported extended pandemic-related stress due to workloads that have increased in proportion to the number of patients, often by 1.5–2 times normal working hours and workloads. Yıldırım and Solmaz [14] have found that such COVID-19-related stress is positively related to elevated COVID-19 burnout.

Previous findings regarding disasters such as SARS have suggested that, during epidemics, all forms of coping measures can alleviate stress and promote mental health [20]. From another angle, a recent systematic review on workplace interventions for employees’ mental health and well-being [21] showed that, regardless of their type, interventions for increasing employees’ control and opportunities for their voices to be heard and for them to participate in the planning and implementation of interventions are critical drivers of well-being. This study therefore explored how employees’ voices and participation during the implementation of COVID-19-oriented actions have been related to burnout. More specifically, this study aimed to determine how COVID-19-specific demands and resources have related to burnout among HCPs. To that aim, it was formulated the following hypotheses:
**Hypothesis** **3.***The COVID-19-specific demands of (a) involvement with COVID-19, (b) fear of COVID-19, and (c) COVID-19-related stress are positively correlated with burnout.*
**Hypothesis** **4.***COVID-19-specific resources such as COVID-19-oriented actions are negatively correlated with burnout.*

Although HCPs were literally applauded around the world for their efforts early during the pandemic, their hero status may have overshadowed the risks to which they have been exposed [22]. As a consequence, HCPs have been liable to feel that they have not been appropriately rewarded for their performance [23], also known as a *breach of the psychological contract*. According to Maslach [23], HCPs are particularly vulnerable to a sense of being insufficiently rewarded, including by being recognized for their work, because positive feedback has been nearly designed out of the process such that a good day at work is often one when nothing bad happens. Studying the fulfillment of the psychological contract among nurses, Sheehan, Tham, Holland, and Cooper [24] have found that unmet expectations in their job are significantly related to nurses’ intention to leave the profession. According to the authors, the importance of job content manifests in consideration of what many nurses describe as a “calling” and the idea that care remains at the heart of the nursing profession—a call that for many has become even stronger during the COVID-19 pandemic.

In the same vein, qualitative research has revealed a strong sense of professional responsibility and identity among HCPs during the COVID-19 pandemic, captured in statements such as “Saving lives is the responsibility of nurses, and I like this profession more” and “Maybe there was discrimination against nurses in the society, but now I am proud of my choice” [17]. Those statements align with Steger, Dik, and Duffy’s [25] definition of *meaningful work* as being both significant and positive in valence (i.e., meaningfulness) and having a eudemonic (i.e., growth- and purpose-oriented) focus. Other research has shown that meaningful work is positively related to well-being, especially meaning in life and life satisfaction, and negatively related to psychological distress, hostility, and depression [25]. Studying mental health personnel in particular, Suarez [26] found that the meaningfulness of work was positively related to all three dimensions of burnout examined—that is, emotional exhaustion, depersonalization, and personal accomplishment.

Thus, this study additionally hypothesized that psychological factors such as meaningful work and breach of the psychological contract relate differently to burnout:
**Hypothesis** **5.***Meaningful work is negatively correlated with burnout.*
**Hypothesis** **6.***Breach of the psychological contract is positively correlated with burnout.*

Because the COVID-19 pandemic has been suggested to affect women with small children the most [27], and because the literature suggests a higher prevalence of burnout among women physicians and nurses [28] and medical workers less than 30 years old [29], whereas working part-time has been found to help doctors to mitigate burnout [30], it was controlled for the variables of gender, age, having children less than 12 years old, and employment status in the analysis.

## 2. Materials and Methods

### 2.1. Study Design and Sample

As part of a larger student project called “Healthy workplaces in light of COVID-19” the present study formed a convenience sample of 268 participants working in health care in Norway. According to Statistics Norway, the corresponding figures for the total of employees in health and social services in Norway was 345,000 in 2019. Data were collected between 25 January and 7 February 2021, at a time when parts of Norway entered into a major social lockdown as the number of COVID-19 infections started to rise again. An online survey using the survey service Nettskjema provided by the University of Oslo was distributed by undergraduate students via email to employees working in either scientific, technical, and administrative services or health and social services. Only respondents who indicated that they were working in health and social services were included in the sample in the present study. The respondents filled out the questionnaire in Norwegian. The scales that were not previously translated into Norwegian was translated and back translated by a bilingual Norwegian-English research assistant and double checked by the researchers involved in the study. All questions in the Nettskjema were required to be answered to fulfill the survey, so there were no missing data. Informed consent was given by clicking “submit” at the end of the survey. The study was conducted in accordance with the guidelines of the Norwegian Center for Research Data.

Of the 268 participants working within health care, 76.5% were women (n = 205) and 23.5% were men (n = 63). The median age group was the age category 41–45 years. Most were spouses or live-in partners with children (44.8%), followed by unmarried individuals without children (28.7%), spouses or live-in partners without children (21.3%), and unmarried individuals with children (5.2%). Of all participants, 18% had children under the age of 12 years, and 53.7% had completed at least 3 years of higher education. Last, by employment status, most had a permanent position (79.1%); 56.7% were working full-time, 24.6% were working a 50–100% position, and 18.7% were working less than a 50% position.

### 2.2. Statistical Analyses

All statistical analyses were performed by using SPSS version 27. The predictive value of organizational, situational (i.e., COVID-19-related), and psychological factors for burnout was tested by conducting 3 × 6 hierarchical regression analyses with the six different burnout dimensions as dependent variables—that is, total BAT score (BAT-Total), the four scores for the four BAT subscales (i.e., BAT Exhaustion, BAT Mental Distance, BAT Cognitive Impairment, and BAT Emotional Impairment), and score on the COVID-19 Burnout Scale (COVID-19-BS). The control variables were entered in a step before the three sets of predictor variables: organizational, situational (i.e., COVID-19-related), and psychological factors. Meanwhile, gender was dummy-coded (1 = woman, 2 = man), as was having children less than 12 years old (0 = children < 12 years old, 1 = children ≥ 12 years old). Age group was an ordinal variable with a 4-year interval starting at 16–20 years, up to 66 years or older, and employment status was coded as 1 (1–50%), 2 (51–99%), or 3 (100%). Inspection of the Q-Q plot, short for quantile-quantile plot, indicated that the residuals for all models followed a normal distribution. Cohen’s ƒ was calculated as a measure of effect size for the multiple regression. Effect size measures for ƒ are 0.02, 0.15, and 0.35, indicating small, medium, and large, respectively. 

### 2.3. Variables

Unless indicated otherwise, all variables were measured on a 5-point Likert scale, with responses ranging from 1 (never) to 5 (always), or “N–A,” or from 1 (totally disagree) to 5 (totally agree), or “D–A.”

#### 2.3.1. Dependent Variables

Burnout was measured with the 12-item short version of the BAT taken from “Manual: Burnout Assessment Tool, Version 2.0” that was developed and validated by Schaufeli, Desart and De Witte [12] and the 10-item COVID-19-BS originally developed by Malach-Pines [13] but adopted to COVID-19 by Yildirim and Solmaz [14]. The BAT (N–A) can be used to assess the presence of burnout as a syndrome (i.e., a 12-item total score, or BAT Total), as well as its four core dimensions, each measured with three items such as “At work, I feel mentally exhausted” (i.e., BAT Exhaustion); “I struggle to find any enthusiasm for my work” (i.e., BAT Mental Distance); “At work, I have trouble staying focused” (i.e., BAT Cognitive Impairment); and “At work, I feel unable to control my emotions” (i.e., BAT Emotional Impairment).

By contrast, the COVID-19-BS (N–A) instrument is an adapted version of the Burnout Measure-Short Version [13] modified by replacing “your work” with “COVID-19” [14]. A sample item is “When you think about COVID-19 overall, how often do you feel hopeless?”

#### 2.3.2. Organizational Factors

Among the organizational factors, work–home conflict (WHC) was measured with a four-item D–A scale developed by Wayne, Musisca, and Fleeson [31] and adapted for use in Norway by Innstrand et al. [32]. One of the items is “Stress at work makes me irritable at home.” By contrast, workload was measured with four N–A items developed by Schaufeli [33]—for example, “Do you have too much work to do?” Next, role conflict was measured with three N–A items also developed by Schaufeli [33], including “Do you get incompatible requests?” After that, autonomy was measured with four D–A items developed by Sverke and Sjöberg [34], including “I can make my own decisions on how to organize my work.” Last, support from colleagues as well as support from leaders was measured with three D–A items each, all also from Schaufeli [33]. Sample item for support from colleagues was “Can you count on your colleagues for help and support, when needed?” Last, sample item for support from leaders was, “Do you feel your work is recognized and appreciated by your supervisor?”

#### 2.3.3. Situational (i.e., COVID-19-Related) Factors

Involvement with COVID-19 was measured with a five-item scale developed by Qian and Li [18] to gauge how intimately individuals have experienced COVID-19. Sample items are “I actively follow the progress of COVID-19” and “I often browse for information on COVID-19 in the news, media, and on the Internet.” Responses were given on a 7-point Likert scale ranging from 1 (totally disagree) to 7 (totally agree). Fear of COVID-19 was measured with a seven-item D–A scale (i.e., FCV-19S) developed by Ahorsu et al. [35] with items such as, “I cannot sleep because I am worrying about getting COVID-19.” COVID-19-related stress was measured with four items from the Perceived Stress Scale [36] adapted for the COVID-19 pandemic by Hayes et al. [37] by asking participants to indicate their thoughts and feelings since the COVID-19 restrictions began. A sample item is “Since the COVID-19 restrictions began, how often have you felt that you were unable to control the important things in your life? Responses were given on a 5-point scale ranging from 1 (very seldom) to 5 (very often). Last, COVID-19-oriented actions were measured with six D–A items developed by Langvik, Karlsen, Saksvik-Lehouillier, and Sorengaard [38] to study work conditions among police during the COVID-19 pandemic. The assessment inquiries into actions, participation, communication, and information provided during the pandemic. A sample item is “The employer has taken good measures to secure the working environment during COVID-19.”

#### 2.3.4. Psychological Factors

Meaningful work was measured with 10 items from the Work and Meaning Inventory [25]. A sample item is “The work I do serves a greater purpose.” The instrument cover three dimensions—positive meaning, meaning-making through work, and greater good motivations—whose scores can be added together to obtain the test-taker’s overall score for meaningful work. Responses were given on a 5-point scale ranging from 1 (totally untrue) to 5 (totally true). Last, breach of the psychological contract (i.e., D–A) was measured by reversing a single item from Schaufeli [33]: “I think that, all in all, I get enough in return for the efforts that I make for my organization.”

## 3. Results

Table 1 provides the means (M), standard deviations (*SD*), Cronbach’s alpha values (α), and correlations between the variables. The mean for COVID-19-BS (1.76) was slightly higher than the mean for BAT Total (1.70) but not as high as the mean for BAT Exhaustion (2.19). The internal consistencies were all satisfactory (α ≥ 70), with Cronbach’s alpha values ranging from 73 to 90. Aside from the intercorrelations between the different burnout measures, BAT Total and BAT Exhaustion were most strongly correlated with WHC, whereas COVID-19-BS was most strongly correlated with the COVID-19 variables (i.e., stress and fear).

Next, Table 2 shows the predictive values of organizational demands and resources on burnout. When controlled for gender, age, having children less than 12 years old, and employment status, WHC was positively and significantly correlated with all burnout measures. Workload was significantly correlated only with BAT Exhaustion and role conflict only with BAT Total and BAT Cognitive Impairment. Autonomy was negatively correlated with BAT Total, BAT Exhaustion, and BAT Mental Distance, whereas support from colleagues was negatively correlated with BAT Total, BAT Exhaustion, and BAT Cognitive Impairment. Support from leaders was related to BAT Exhaustion but not negatively as hypothesized. The model explained 12–48% of the variance in the different burnout measures. The effect size (Cohen’s *f*) was large for all models.

With the background variables controlled for, COVID-19 involvement was positively correlated only to COVID-19-related burnout (i.e., COVID-19-BS). By contrast, fear of COVID-19 was positively correlated to all burnout measures except BAT Mental Distance and BAT Cognitive Impairment. However, COVID-19-related stress, the strongest predictor of all dependent variables, was positively correlated to all burnout measures. Last, COVID-19-oriented actions was negatively correlated only to BAT Mental Distance. The model explained 12–52% of the variance in the different burnout measures (see Table 3). The effect size (Cohen’s *f*) was large for all models.

Additionally, with the background variables controlled for, meaningful work was negatively correlated with BAT Total and BAT Mental Distance, whereas breach of the psychological contract was positively correlated with all variables. The model explained 3–23% of the variance in the different burnout measures (see Table 4). The effect size (Cohen’s *f*) was large for BAT Total, BAT Exhaustion, BAT Mental Distance, and COVID-9-BS, but medium for BAT Cognitive Impairment and BAT Emotional Impairment.

## 4. Discussion

In this study, we sought to understand the high prevalence of burnout among HCPs during the COVID-19 pandemic by investigating burnout’s organizational, situational (i.e., COVID-19-related), and psychological triggers.

In line with findings from before the COVID-19 pandemic, in this study WHC was positively associated with higher levels of burnout, which supported Hypothesis 1(a). The strong observed effect of WHC on burnout is consistent with the results of a longitudinal study in Norway that suggest a reciprocal relationship between those two variables for different occupational groups [39]. Exploring turnover intention among nurses in New Zealand, Moloney [8] also identified workload and work–life interference as the strongest predictors of burnout and, in turn, the intention to leave the profession. However, in the present study, workload was related only to BAT Exhaustion, which partly supported Hypothesis 1(b). Nevertheless, these results concur with recent findings showing that feeling pushed beyond training (i.e., high workload) during the pandemic has been positively related to emotional exhaustion among HCPs in 60 countries [2]. Meanwhile, role conflict was positively correlated with BAT Total and BAT Cognitive Impairment, which partly supported Hypothesis 1(c). The exhausting nature of workload makes sense in that being assigned incompatible tasks (i.e., role conflict) induces increased cognitive depletion. More broadly, the results suggest that using the different subscales of burnout provides a more nuanced picture of burnout and its predictors [12].

Hypothesis 2(a) was only partly supported because autonomy was negatively related to BAT Total, BAT Exhaustion, and BAT Mental Distance but not BAT Cognitive Impairment, BAT Emotional Impairment, or COVID-19-related burnout (i.e., COVID-19 BS). These negative relationships with burnout align with past results, suggesting that workplace flexibility mitigates the negative effects of burnout among HCPs [40].

Support from colleagues was negatively related to BAT Total, BAT Exhaustion, and BAT Cognitive Impairment, which partly supported Hypothesis 2(b). Thus, support from colleagues seems able to mitigate exhaustion and to sustain focus. By contrast, support from leaders was related only with BAT Exhaustion but not negatively as hypothesized. Because the bivariate correlations between support from leaders and BAT Exhaustion were negative (r = −20), the model was tested without support from colleagues, which revealed a negative sign that could indicate a suppressive effect. The organizational factors, particularly WHC, explained most of the variance for BAT Total and BAT Exhaustion.

Hypothesis 3(a) was only partly supported because involvement with COVID-19 was only positively related to COVID-19 burnout (i.e., COVID-19 BS) and not general burnout (i.e., BAT Total). Thus, actively following the progress of the COVID-19 pandemic and browsing for information about COVID-19 in the news, media, and online might contribute to COVID-19 burnout. The fact that that situational factor was related only to COVID-19-specific burnout indicates that pandemic-specific assessment tools might be useful for fully understanding the problems and mental challenges that can arise during a pandemic [14]. Previous research has shown that involvement with COVID-19 was positively correlated with mental health conditions such as fear and anxiety [18]. This study’s results add to that finding by relating involvement with COVID-19 to COVID-19 burnout.

Fear of COVID-19, on the contrary, was positively related to all burnout measures except BAT Mental Distance and BAT Cognitive Impairment. The non-significant relationship with BAT Mental Distance and BAT Cognitive Impairment was surprising, because fear can be viewed as a mental and a cognitive state. Previous studies have shown that feelings such as fear and anxiety among HCPs have seemingly decreased since the COVID-19 pandemic peaked or once the employees had adapted to their new workplace situations [3]. Thus, the non-significant relationship of COVID-19 burnout with mental and cognitive impairment could be because adapting to the situation has not required great mental or cognitive energy yet, but rather fatigue. Nevertheless, similar to involvement with COVID-19, this situational factor was most strongly related to COVID-19 burnout.

Overall, COVID-19-related stress was the strongest predictor among all dependent situational variables and positively correlated with all burnout measures, which supported Hypothesis 3(c). Ample literature suggests a positive association between work-related stress and burnout not only in general [41] but also among HCPs during the COVID-19 pandemic [2]. In line with Yıldırım and Solmaz’s [14] findings, these results also suggest that COVID-19-related stress is related to COVID-19 burnout. As mentioned, whereas workload did not predict COVID-19 burnout, COVID-19-related stress seems to be a strong predictor of such burnout.

It was also hypothesized that COVID-19-oriented actions, participation, communication, and information provided during the pandemic would mitigate burnout. That hypothesis, H4, was only partly supported because the variable, COVID-19-oriented actions, was related only to BAT Mental Distance. Even so, that finding corroborates Demerouti et al.’s [15] results suggesting that on-the-job resources are primarily related to disengagement and/or depersonalization (i.e., mental distance) because such resources may inversely influence mental distance by minimizing or reducing its use as a coping strategy [11]. According to Schaufeli et al. [12], BAT Mental Distance differs from the other subscales for burnout (i.e., BAT Exhaustion, BAT Cognitive Impairment, and BAT Emotional Impairment) by capturing an unwillingness to invest energy, whereas the other subscales capture the inability to invest energy. Thus, it seems that actions initiated by an organization have a relatively strong mitigating effect on employees’ unwillingness to invest energy in their work. Although that hypothesis makes sense, it needs to be more thoroughly investigated. If true, then providing a good communication channel and informing employees about strategies in response to pandemics could be beneficial for reducing mental distance among HCPs. Overall, the situational factors (i.e., COVID-19-related factors) explained far more of the explained variance in the COVID-19 burnout measure (52%) than the BAT measures (12–32%).

Hypothesis 4, that meaningful work would be negatively correlated with burnout, was partly supported because it was significantly related to BAT Total, BAT Exhaustion, and BAT Mental Distance. Those results align with past findings from studies on HCPs showing that meaningful work was positively correlated with all three dimensions of burnout—that is, emotional exhaustion, depersonalization, and personal accomplishment [26]. Personal accomplishment was not measured in the present study but instead assessed cognitive and emotional impairment, neither of which was related to meaningful work. Meaningful work was also not related to COVID-19-BS. The somewhat stronger relationship between meaningful work and BAT Mental Distance than with BAT Exhaustion supports the predictions of the JD R model [11] as well as previous findings suggesting that psychological job resources such as meaningfulness of work are primarily related to the mental aspect of burnout [15].

A breach of the psychological contract, by contrast, was positively related to all burnout measures. Thus, HCPs’ feeling that they have not received sufficient reward for the efforts that they have made for their organizations during the COVID-19 pandemic seems to be devastating for their mental health. That reasoning concurs with social exchange theory’s suggestion that burnout occurs when workers perceive a lack of equity between their efforts and contributions made at work and the results obtained [11]. According to Yuan et al. [22], the hero status given to HCPs worldwide early during the pandemic has not compensated for the risks to which they have exposed themselves and their families. In Norway, as in many countries, wages for HCPs working in lower positions have been debated since the pandemic’s outbreak; however, a breach of the psychological contract can also relate to a lack of acknowledgment from leaders, colleagues, patients, and/or society at large. Previous studies have linked the fulfillment of the psychological contract to an increase in turnover intention among nurses [24]. The present study add to that finding by suggesting that breaches of the psychological contract are associated with higher levels of burnout among HCPs. In response, strategies that narrow the gap between employees’ expectations and their return on investment from their organizations seems to benefit both employees’ mental health and the turnover intention and reduced disorder-related costs at their organizations. Taken together, those two psychological factors explained 3–23% of the variance in the various burnout measures.

### 4.1. Implications

Overall, the results suggest that COVID-19 burnout can be better explained by situational (i.e., COVID-19-related) factors than organizational or psychological factors. That dynamic implies that pandemic-specific tools may help to clarify factors that affect the mental health of employees, as suggested by Yıldırım and Solmaz [14].

In line with a general consensus in the literature that exhaustion is the essential dimension of burnout [12], these results indicate that most of the variables indeed seem to be related to that dimension. In fact, BAT Exhaustion had the highest mean. Even so, the different relationships between the different subscales for burnout indicate that using a more refined measure of burnout can afford a more nuanced picture of its antecedents. In line with the JD R framework, job demands seem to be most related to BAT Exhaustion and on-the-job resources most related to BAT Mental Distance. Thus, to prevent burnout, organizations should try to reduce demands and facilitate resources for their employees.

One of the strongest predictors of burnout in general seems to be WHC. That result aligns with findings from before the pandemic in general [12] and among women physicians in particular [42]. However, WHC might have been exceptionally challenging for employees during the COVID-19 pandemic as they have attempted to balance new demands at work with sickness, fear for their family members, and home schooling. Organizations need to acknowledge that their employees’ health and well-being are not exclusively determined by factors at work and to find ways of easing the imbalance between work and family life. Given predicted shortages of nurses in the future [8] and Generation Y nurses’ (i.e., born between 1980 and 1994) strong desire for work–life balance [43], any action that can balance the needs for service delivery with the needs of staff is highly recommended. Giusino et al.’s findings [44] suggests that HCP’s mental health should be addressed on multiple levels (individual, the group, the leader, and the organization) as COVID-19 related job demands and resources are found at all levels [45].

Last, in addition to reducing stress and workload, easing WHC, and facilitating meaningful work, autonomy, and support from colleagues, organizations should pay attention to the psychological contract that they have struck with HCPs. Balancing the requirements of the organization with employees’ expectations seems essential not only for the health and well-being of the employees but also the services that they provide, which has implications for patients and society.

### 4.2. Limitations

This article contributes to the literature by exploring organizational, situational, and psychological triggers of burnout in general and COVID-19 burnout at a time when HCPs in Norway faced unusual pressure. Although these findings offer unique insights into an extreme situation, they should also be interpreted and generalized with that circumstance of this study in mind. Even though it was used different types of burnout assessments, the results are all based on self-report assessments. For an alternative, future studies should use implicit measures or behavioral indicators alongside self-report questionnaires while studying the relationship between different triggers and burnout. Among other limitations, causal interpretations cannot be made based on the study’s cross-sectional design, and the data were collected using non-representative sampling. Although utilizing a snowball sampling method provided convenience for our data collection in a time where we needed to respond quickly to gather valuable data, this nonrandom sampling might have provided a selection bias that undermines the representativeness and generalization of the results. Moreover, in the present study, men were underrepresented. Although the somewhat lower participation of men reflects what international studies conducted in health care have also experienced [3], the results are nevertheless based on a sample in which women are overrepresented.

## 5. Conclusions

Altogether, the present study revealed that WHC, COVID-19-related stress, and breach of the psychological contract are important contributors to all burnout measures and that using different subscales for burnout provides a more nuanced picture of its antecedents. Moreover, the findings suggest that exhaustion plays a central role in burnout, that pandemic-specific assessment tools are needed to clarify how mental health is affected during pandemics, and that both organizational and situational factors are differently related to general burnout and COVID-19 burnout. All of these aspects have implications for future research, assessment, and practice geared toward mitigating burnout among HPCs.

## Figures and Tables

**Table 1 ijerph-19-11807-t001:** Mean (M), standard deviation (SD), correlation (two-tailed Pearson’s r), and Cronbach’s alpha for the study variables.

	M	*SD*	1	2	3	4	5	6	7	8	9	10	11	12	13	14	15	16	17
1	1.7	0.5	0.86																
2	2.2	0.8	0.81	0.86															
3	1.6	0.7	0.81	0.56	0.73														
4	1.7	0.5	0.68	0.34	0.38	0.77													
5	1.3	0.4	0.70	0.38	0.46	0.48	0.75												
6	1.8	0.7	0.63	0.57	0.50	0.39	0.40	0.90											
7	2	0.7	0.31	0.31	0.16	0.21	0.24	0.42	0.84										
8	2.4	0.8	0.53	0.49	0.38	0.35	0.34	0.62	0.36	0.78									
9	5.7	1	0.14	0.11 ^ns^	0.12	0.10 ^ns^	0.07 ^ns^	0.31	0.13	0.10 ^ns^	0.77								
10	3.7	0.7	−0.21	−0.13	−0.26	−0.15	−0.07 ^ns^	−0.13	0.00 ^ns^	−0.24	0.06 ^ns^	0.79							
11	3.0	0.9	0.55	0.58	0.36	0.31	0.33	0.40	0.35	0.39	0.13	−0.14	0.81						
12	3.4	0.7	0.27	0.32	0.15	0.16	0.15	0.13	0.12	0.07 ^ns^	0.14	0.02 ^ns^	0.40	0.80					
13	2.6	0.7	0.35	0.30	0.26	0.27	0.22	0.14	0.12	0.12	0.13	−0.22	0.36	0.51	0.75				
14	3.6	0.8	−0.36	−0.34	−0.37	−0.15	−0.15	−0.24	−0.10 ^ns^	−0.31	−0.07 ^ns^	0.37	−0.21	−0.05 ^ns^	−0.23	0.85			
15	4	0.6	−0.32	−0.33	−0.26	−0.23	−0.07 ^ns^	−0.11 ^ns^	−0.15	−0.21	0.04 ^ns^	0.27	−0.33	−0.09 ^ns^	−0.22	0.25	0.76		
16	3.7	0.9	−0.22	−0.20	−0.20	−0.13	−0.12	−0.12	−0.13	−0.20	0.02 ^ns^	0.45	−0.25	−0.03 ^ns^	−0.28	0.41	0.49	0.87	
17	3.9	0.5	−0.18	−0.10 ^ns^	−0.32	−0.11 ^ns^	0.02 ^ns^	−0.08 ^ns^	−0.07 ^ns^	−0.10 ^ns^	0.14	0.26	−0.04 ^ns^	0.02 ^ns^	−0.04 ^ns^	0.30	0.35	0.23	0.80
18	2.6	0.9	0.39	0.38	0.38	0.17	0.19	0.23	0.02 ^ns^	0.24	0.03 ^ns^	−0.44	0.29	0.14	0.32	−0.47	−0.33	−0.45	−0.21

Note: All significant at *p* < 0.05 if not marked with ^ns^ = not significant. Cronbach’s alpha in italics at the diagonal 1. BAT Burnout Total; 2. BAT Exhaustion; 3. BAT Mental Distance; 4. BAT Cognitive Impairment; 5. BAT Emotional Impairment; 6. Burnout COVID-19; 7. Fear of COVID-19; 8. Stress during COVID-19; 9. COVID-19 Involvement; 10. COVID-19 Actions; 11. Work-Home Conflict (WHC); 12. Workload; 13. Role Conflict; 14. Autonomy; 15. Colleague Support; 16. Leader Support; 17. Meaningful work; 18. Psychological contract breach.

**Table 2 ijerph-19-11807-t002:** Linear regression analyses for the organizational factors on the different burnout measures (BAT and COVID-19-BS).

	BAT Total		BAT Exhaustion		BAT Mental Distance		BAT Cognitive Impairment		BAT Emotional Impairment		COVID-19-BS	
	β	95% CI	β	95% CI	β	95% CI	β	95% CI	β	95% CI	β	95% CI
*Background variables:*												
Gender	0.06	−0.05/0.17	0.05	−0.08/0.27	0.07	−0.07/0.28	0.04	−0.10/0.20	0.02	−0.14/0.11	−0.03	−0.22/0.13
Age	−0.22 ***	−0.06/−0.02	−0.27 ***	−0.11/−0.05	−0.16 *	−0.07/−0.01	−0.11	−0.05/0.01	−0.06	−0.03/0.01	−0.30 ***	−0.10/−0.04
Children < 12	0.15 **	0.06/0.39	0.17 **	0.17/0.53	0.07	−0.07/0.30	0.06	−0.07/0.25	0.14	0.02/0.28	0.14 *	−0.05/0.41
Employment status	0.10	−0.02/0.12	0.07	−0.04/0.17	0.07	−0.05/0.16	0.05	−0.06/0.12	0.13	−0.00/0.13	−0.02	−0.12/0.09

∆*R*^2^		*0.09 ****		*0.13 ****		*0.05 ***		*0.02*		*0.04*		*0.16 ****

*Demands:*												
WHC	0.40 ***	0.15/0.27	0.41 ***	0.28/0.48	0.25 ***	0.09/0.29	0.21 **	−0.05/0.22	0.31 ***	0.08/0.22	0.33 ***	0.16/0.35
Workload	0.05	−0.05/0.12	0.16 **	0.05/0.32	−0.00	−0.13/0.13	−0.01	−0.12/0.11	−0.05	−0.13/0.06	0.04	−0.09/0.17
Role conflict	0.14 *	0.02/0.18	0.04	−0.09/0.18	0.12	−0.01/0.25	0.18 *	0.03/0.26	0.13	−0.01/0.18	−0.00	−0.14/0.17

∆*R*^2^		*0.30 ****		*0.32 ****		*0.15 ****		*0.12 ****		*0.11 ****		*0.11 ****

*Resources:*												
Autonomy	−0.22 ***	0.20/−0.07	−0.19 ***	−0.31/−0.10	−0.28 ***	−0.35/−0.13	−0.05	−0.13/0.06	−0.10	−0.14/0.02	−0.11	−0.21/0.01
Colleague support	−0.15 **	−0.19/−0.03	−0.19 ***	−0.36/−0.10	−0.13	−0.27/−0.01	−0.15 *	−0.24/−0.01	0.09	−0.03/0.15	−0.00	−0.14/0.13
Leader support	0.11	−0.00/0.12	0.13 *	0.02/0.21	0.09	−0.03/0.16	0.07	−0.04/0.13	0.00	−0.07/0.07	0.06	−0.06/0.14
∆*R*^2^		*0.05 ****		*0.05 ****		*0.07 ****		*0.02*		*0.01*		*0.01*
*R^2^(adjusted)*		*0.42*		*0.48*		*0.24*		*0.12*		*0.12*		*0.26*
Cohen’s *f*		*0.89*		*0.99*		*0.61*		*0.43*		*0.43*		*0.63*

*NOTE*: BAT: Burnout Assessment Tool; COVID-19-BS: COVID-19 Burnout Scale; Children < 12: Children under the age of 12 years; β: Standardized beta; 95% CI: 95% Confidence Interval for B; *R^2^*: adjusted R-square; ∆*R*^2^: R-square change; Cohen’s *f*: effect size; * *p* < 0.05, ** *p* < 0.01, *** *p* < 0.001.

**Table 3 ijerph-19-11807-t003:** Linear regression analyses for the situational (COVID-19) factors on the different burnout measures (BAT and COVID-19-BS).

	BAT Total		BAT Exhaustion		BAT Mental Distance		BAT Cognitive Impairment		BAT Emotional Impairment		COVID-19 BS	
	β	95% CI	β	95% CI	β	95% CI	β	95% CI	β	95% CI	β	95% CI
*Background variables:*												
Gender	−0.06	−0.18/0.04	−0.08	−0.35/0.04	−0.01	−0.18/0.16	−0.01	−0.16/0.13	−0.09	−0.21/0.02	−0.10 *	−0.29/−0.02
Age	−0.14 *	−0.04/−0.00	−0.21 ***	−0.10/−0.03	−0.09	−0.05/0.01	−0.03	−0.03/0.02	−0.05	−0.03/0.01	−0.23 ***	−0.08/−0.03
Children < 12	0.12 *	0.02/0.26	0.16 **	0.10/0.52	0.05	−0.11/0.26	0.03	−0.12/0.20	0.10	−0.02/0.24	00.08	−0.01/0.28
Employment status	0.10	−0.01/0.12	0.08	−0.04/0.19	0.03	−0.08/0.13	0.07	−0.04/0.13	0.14 *	−0.01/0.15	−0.00	−0.08/0.08

∆*R*^2^		*0.09 ****		*0.13 ****		*0.05 **		*0.02*		*0.04 **		*0.16 ****

*C−19 Demands:*												
COVID−19 Inv.	0.07	−0.01/0.08	0.04	−0.06/0.11	0.10	−0.01/0.14	0.06	−0.03/0.10	0.02	−0.04/0.06	0.22 ***	0.09/0.21
COVID−19 Fear	0.15 **	0.03/0.17	0.17 **	0.06/0.31	0.05	−0.07/0.16	0.11	−0.02/0.17	0.13 *	0.00/0.15	0.21 ***	0.11/0.29
COVID−19 Stress	0.41 ***	0.18/0.32	0.36 ***	0.25/0.50	0.29 ***	0.14/0.36	0.28 ***	0.11/0.29	0.29 ***	0.09/0.24	0.45 ***	0.31/0.48


∆*R*^2^		*0.24 ****		*0.19 ****		*0.12 ****		*0.12 ****		*0.12 ****		*0.37 ****

*C−19 resources:*												
COVID−19 Act.	−0.10	−0.13/0.01	−0.01	−0.13/0.11	−0.18 **	−0.27/−0.06	−0.10	−0.16/0.02	−0.02	−0.08/0.06	0.02	−0.07/0.10

∆*R*^2^		*0.01*		*0.00*		*0.03 ***		*0.01*		*0.00*		*0.00*
*R* ^2^ *(adjusted)*		*0.32*		*0.30*		*0.17*		*0.12*		*0.13*		*0.52*
Cohen’s *f*		*0.71*		*0.68*		*0.49*		*0.41*		*0.44*		*1.06*

*NOTE:* BAT: Burnout Assessment Tool; COVID-19-BS: COVID-19 Burnout Scale; Children < 12: Children under the age of 12 years; β: Standardized beta; 95% CI: 95% Confidence Interval for B; *R*^2^: adjusted R-square; ∆*R*^2^: R-square change; Cohen’s *f*: effect size; COVID-19 Inv: COVID-19 Involvement; COVID-19 Act: COVID-19 Actions. * *p* < 0.05, ** *p* < 0.01, *** *p* < 0.001.

**Table 4 ijerph-19-11807-t004:** Linear regression analyses for the psychological factors on the different burnout measures (BAT and COVID-19-BS).

	BAT Total		BAT Exhaustion		BAT Mental Distance		BAT Cognitive Impairment		BAT Emotional Impairment		COVID-19-BS	
	β	95% CI	β	95% CI	β	95% CI	β	95% CI	β	95% CI	β	95% CI
*Background variables:*												
Gender	−0.06	−0.18/0.05	−0.07	−0.34/0.06	−0.01	−0.17/0.16	−0.02	−0.18/0.12	−0.09	−0.21/0.03	−0.12 *	−0.37/−0.02
Age	−0.24 ***	−0.06/−0.02	−0.27 ***	−0.11/−0.04	−0.18 **	−0.07/−0.01	−0.11	−0.05/0.00	−0.10	−0.04/0.01	−0.34 ***	−0.11/−0.05
Children < 12	0.18 ***	0.08/0.34	0.21 ***	0.21/0.64	0.10	−0.01/0.35	0.08	−0.06/0.27	0.14 *	0.02/0.28	0.16 **	0.07/0.45
Employment status	0.06	−0.04/0.10	0.04	−0.08/0.16	0.00	−0.10/0.10	0.04	−0.06/0.12	0.11	−0.02/0.13	−0.03	−0.13/0.05

∆*R*^2^		*0.09 ****		*0.13 ****		*0.05 **		*0.02*		*0.04 **		*0.16 ***

Meaningful work	−0.14 *	−0.23/−0.02	−0.06	−0.26/0.08	−0.27 ***	−0.49/−0.21	−0.09	−0.23/0.03	0.04	−0.07/0.14	−0.08	−0.26/0.05

∆*R*^2^		*0.04 ****		*0.02 **		*0.11 ****		*0.02 **		*0.00*		*0.01 **

Psychological contract breach	0.33 ***	0.11/0.23	0.33 ***	0.20/0.40	0.30 ***	0.14/0.31	0.13 *	0.01/0.16	0.18 **	0.03/0.15	0.15 **	0.03/0.20

∆*R*^2^		*0.10 ****		*0.10 ****		*0.08 ****		*0.02 **		*0.03 ***		*0.02 ***
*R* ^2^ *(adjusted)*		*0.22*		*0.23*		*0.22*		*0.03*		*0.05*		*0.18*
Cohen’s *f*		*0.55*		*0.57*		*0.56*		*0.23*		*0.27*		*0.49*

*NOTE:* BAT: Burnout Assessment Tool; COVID-19-BS: COVID-19 Burnout Scale; Children < 12: Children under the age of 12 years; β: Standardized beta; 95% CI: 95% Confidence Interval for B; *R*^2^: adjusted R-square; ∆*R*^2^: R-square change; Cohen’s *f*: effect size. * *p* < 0.05, ** *p* < 0.01, *** *p* < 0.001.

## Data Availability

The data presented in this study are available on request given to the corresponding author. The data are not publicly available due to privacy reasons.

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
