# Peer review of "Burnout among Health Care Professionals during COVID-19"

_ijerph, 2022, doi:10.3390/ijerph191811807_

Round 1

Reviewer 1 Report

This paper investigated the effect of organizational, situational, and psychological factors on burnout during the COVID-19 pandemic among 268 health care professionals in Norway.   The modeling results showed how the pandemic has affected mental health.  The conclusions look reasonable and the paper is well organized.  I have a few comments hope can improve the paper.

1. For the survey data collection, how did the author control the selection bias? Because the online survey may be answered by certain group of people, which may give biased inference without controlling certain important variables (ie. omit variables) in the statistical models.

2. Are there missing data and how the missing data were handled? 

3. In table 1, the means and standard deviations were both provided for continuous and categorical variables.  But for categorical variables, the proportions are expected, rather than the means.  

4. For the statistical models in table 2 and 3, as age and employment status  were discretized into levels, why there is only one \beta for each variables?  By this way, they were treated as continuous variables in the model.  If they are categorical as described, there suppose to be (number of total level - 1) \beta's for each variables.

Author Response

This paper investigated the effect of organizational, situational, and psychological factors on burnout during the COVID-19 pandemic among 268 health care professionals in Norway.   The modeling results showed how the pandemic has affected mental health.  The conclusions look reasonable and the paper is well organized.  I have a few comments hope can improve the paper.

Thank you for the comments for improvement, and the possibility to revise and resubmit the paper. Responses are given in red bellow.

  1. For the survey data collection, how did the author control the selection bias? Because the online survey may be answered by certain group of people, which may give biased inference without controlling certain important variables (ie. omit variables) in the statistical models.

Response: Unfortunately, using a convenience sample we were not able to control for selection bias precluding any generalization of the findings. This selection bias is highlighted more in the limitation section now

  1. Are there missing data and how the missing data were handled? 

Response: All questions in the Nettskjema were required to be answered to fulfill the survey, so there were no missing in the data. This information is added to the manuscript.

  1. In table 1, the means and standard deviations were both provided for continuous and categorical variables.  But for categorical variables, the proportions are expected, rather than the means.

Response: Table 1 consists of continuous variables only, hence their mean and SD are provided.

  1. For the statistical models in table 2 and 3, as age and employment status  were discretized into levels, why there is only one \beta for each variables?  By this way, they were treated as continuous variables in the model.  If they are categorical as described, there suppose to be (number of total level - 1) \beta’s for each variables.

Response: By levels, do you mean that they are categorial? In case, all background variables are categorial. How they are coded is described in the method section.

“Meanwhile, gender was dummy-coded (1 = woman, 2 = man), as was having children less than 12 years old (0 = children <12 years old, 1 = children ≥12 years old). Age group was a categorical variable with a 4-year interval starting at 16–20 years, up to 66 years or older, and employment status was coded as 1 (1–50%), 2 (51–99%), or 3 (100%).”

This implies that for example a significant positive beta of gender on BAT Total, implies more burnout among men. However, these background variables are only used as control variables and not part of the hypotheses

Reviewer 2 Report

Dear Author,

Congratulations for the work done. This is interesting research but it needs to be improved to be accepted in this Journal.

ABSTRACT: The author uses "our" although only one author appears in the manuscript. In any case, the use of the first person would be discouraged. There is an excess and abuse of acronyms and abbreviations in this section (in which their simple use is discouraged).

INTRODUCTION: This section is correct although it should be complemented with the use of more and recent bibliographic references on the impact of confinement (https://www.mdpi.com/search?q=covid+teleworking&journal=ijerph) and burnout on the different tasks and skills of healthcare professionals (doi: 10.1080/07853890.2022.2059102).

METHODS: This section is clearly deficient, lacking many data about the procedure followed to obtain the sample, the calculation of its statistical power, to obtain the data and their treatment. I advise the author to consult the PERSIST checklist corresponding to the study design of the manuscript to transmit all the necessary information in this section to ensure the reliability, validity and reproducibility of the research. In addition, you should also attend to the order in which such information should be provided.

RESULTS: Table 1 is counterintuitive and should be redesigned to facilitate interpretation. It should provide the 95% confidence intervals, at least, of the Bs of the regression analyses. Conveying a decimal value of the mean and standard deviation values is sufficient, what added meaning does the second decimal value bring? The zeros as the last decimal place should be eliminated, what information does it provide? The analyses performed should be complemented with the calculation of the effect sizes.

DISCUSSION: It is solid and coherent although it would also benefit from some new references already recommended in the Introduction section.

Kind regards.

Author Response

Dear Author,

Congratulations for the work done. This is interesting research but it needs to be improved to be accepted in this Journal.

Thank you for the comments for improvement, and the possibility to revise and resubmit the paper. Responses are given in red bellow.

ABSTRACT: The author uses “our” although only one author appears in the manuscript. In any case, the use of the first person would be discouraged. There is an excess and abuse of acronyms and abbreviations in this section (in which their simple use is discouraged).

Response: Thank you for notifying this. Unfortunately, this is a result after a professional language editing service, which was not aware that it was only one author and who wanted a more active language. I have gone through the whole paper and replaced any “our” or “we” with “this study” etc.

I see that the abbreviation BAT is used several times in the abstract, this is now removed from the description of the BAT dimensions.

INTRODUCTION: This section is correct although it should be complemented with the use of more and recent bibliographic references on the impact of confinement (https://www.mdpi.com/search?q=covid+teleworking&journal=ijerph) and burnout on the different tasks and skills of healthcare professionals (doi: 10.1080/07853890.2022.2059102).

Response: Due to the theme of COVID-19 most of the studies are new and from the last three years. In fact, out of the 30 references provided in the introduction 16 are from 2020-2022, and 5 are from 2018 and 2019. The rest of the references are related to original definitions and theories. However, I can see that the reference list was missing in this review version, and that it could have been difficult to evaluate. I have added this list again so it easier for you to see the literature included. Unfortunately, the references suggested are basically on telework which is not a part of this study (https://www.mdpi.com/search?q=covid+teleworking&journal=ijerph). The last reference could have been included (doi: 10.1080/07853890.2022.2059102) as it suggests that burnout is related to lower empathy in physiotherapists. However, as reviewer 3 suggest the introduction part to be shorter, and not longer, and it is included a review from 2021* stating the effect of burnout on patients already this reference did not add any new, substantial information to the paper. However, I have added some more references in the discussion as suggested by you

*Jun, J.;  Ojemeni, M. M.;  Kalamani, R.;  Tong, J.; Crecelius, M. L., Relationship between nurse burnout, patient and organizational outcomes: Systematic review. In-ternational Journal of Nursing Studies 2021, 119, 103933.

METHODS: This section is clearly deficient, lacking many data about the procedure followed to obtain the sample, the calculation of its statistical power, to obtain the data and their treatment. I advise the author to consult the PERSIST checklist corresponding to the study design of the manuscript to transmit all the necessary information in this section to ensure the reliability, validity and reproducibility of the research. In addition, you should also attend to the order in which such information should be provided.

Response: This section now provides more information about the collection of data. The use of convenience sample and a snowball sampling method to gather the data precludes any generalization. This is elaborated more in the limitation section as well. It also makes it hard to calculate any statistical power of the sample. Instead, I have added the corresponding figures for the total of employees in health and social services in Norway

RESULTS: Table 1 is counterintuitive and should be redesigned to facilitate interpretation. It should provide the 95% confidence intervals, at least, of the Bs of the regression analyses. Conveying a decimal value of the mean and standard deviation values is sufficient, what added meaning does the second decimal value bring? The zeros as the last decimal place should be eliminated, what information does it provide? The analyses performed should be complemented with the calculation of the effect sizes.

Response: I have changed all Means and SD in table 1 to one decimal as suggested.  I have also redone all regression analysis and added 95% Confidence interval for beta (B) and calculated Cohen’s d effect size. The text is updated accordingly

DISCUSSION: It is solid and coherent although it would also benefit from some new references already recommended in the Introduction section

Thank you. We have added these two new references in the discussion. Both are from 2022.

Giusino, D.;  De Angelis, M.;  Mazzetti, G.;  Faiulo, I.;  Innstrand, S. T.;  Christensen, M.; Nielsen, K., Mentally Healthy Healthcare: Main Findings and Lessons Learned From a Needs Assessment Exercise at Multiple Workplace Levels. 2022.

Giusino, D.;  De Angelis, M.;  Mazzetti, G.;  Christensen, M.;  Innstrand, S. T.;  Faiulo, I. R.; Chiesa, R., "We All Held Our Own": Job Demands and Resources at In-dividual, Leader, Group, and Organizational Levels During COVID-19 Outbreak in Health Care. A Multi-Source Qualitative Study. Workplace Health Saf 2022, 70 (1), 6-16.

Reviewer 3 Report

The topic of the work is interesting, but the realization is flawed.

1. First of all, only one author is listed on the authors list, but in the manuscript I can see the phrases "our study", "we". It is strange and implicate that more than one author participated in the study.

2. I do not prefer the term "mental illness". It is too strong. In this type of study it is better to use the term "mental disorders".

3. Lines 56, 57, 58 --> references needed.

4. Introduction is too long. Need to be shorter, and substantiality and summarized.

5. The sample size needs to be calculated, based on the full number of health care professionals in Norway.

6. Who approved this study? What Ethical Committee? Need to be indicated.

7. How did the health care professionals fill out this questionnaire? In English or in Norwegian? If in Norwegian who translates this questionnaire. Who validates questionnaires? What are the croblach alfa for each guastionnares? It is important to have a final Cronbalch alpha for each questionnaire, not each question.

8. How was the distribution of the data? Normal? Not normal. According to that calculate mean and SD or Median and interquartile.

9. Tables 2, 3 and 4 there are completely incomprehensible. If you calculate regression analysis you need to show beta and P, or even better make a graph with T value.

10. The names of tables 2, 3 and 4 do not correspond to what I see in them.

11. You need to have less hypothesis. In fact only one!

Author Response

Thank you for the comments for improvement, and the possibility to revise and resubmit the paper. The responses are given in red bellow.

The topic of the work is interesting, but the realization is flawed.

  1. First of all, only one author is listed on the authors list, but in the manuscript I can see the phrases “our study”, “we”. It is strange and implicate that more than one author participated in the study.

Response: Thank you for notifying this. Unfortunately, this is a result after a professional language editing service, which was not aware that it was only one author and who wanted a more active language. I have gone through the whole paper and replaced any “our” or “we” with “this study” etc.

  1. I do not prefer the term “mental illness”. It is too strong. In this type of study it is better to use the term “mental disorders”.

 Response: I agree and have replaced any “mental illness” with “mental disorder”.

  1. Lines 56, 57, 58 --> references needed.

Response: This was my statement only.  However, this statement is removed  from the text now

  1. Introduction is too long. Need to be shorter, and substantiality and summarized.

Response: The introduction section has been shortened to make the text more substantiality and summarized.

  1. The sample size needs to be calculated, based on the full number of health care professionals in Norway.

Response: According to Statistics Norway the corresponding figures for the total of employees in health and social services in Norway were 345 000 in 2019. This have been added to the method section.

  1. Who approved this study? What Ethical Committee? Need to be indicated.

 Response: By using Nettskjema and keeping background information to a minimum the present study was in accordance with the guidelines of the Norwegian Center for Research Data, hence no additional ethical approval was not required. This is clarified in the manuscript now

  1. How did the health care professionals fill out this questionnaire? In English or in Norwegian? If in Norwegian who translates this questionnaire. Who validates questionnaires? What are the croblach alfa for each guastionnares? It is important to have a final Cronbalch alpha for each questionnaire, not each question.

Response: The health care professionals filled out the questionnaire in Norwegian. The scales that were not previously translated into Norwegian was translated and back translated by a bilingual Norwegian English research assistant and double checked by the researchers involved in the study. This information is now added in the method section.  Cronbach’s alpha for all study variables is provided in italics at the diagonal in the Table 1.

  1. How was the distribution of the data? Normal? Not normal. According to that calculate mean and SD or Median and interquartile.

Response: In regression analyses there is an assumption that the residuals needs to be normal, and not the DV or IV. Thus, I have added this to the text: "Inspection of the Q-Q plot, short for quantile-quantile plot, indicated that the residuals for all models followed a normal distribution. "

  1. Tables 2, 3 and 4 there are completely incomprehensible. If you calculate regression analysis you need to show beta and P, or even better make a graph with T value.

Response: As suggested also by reviewer 2 I have redone all regression analysis and added 95% Confidence interval for beta (B) instead and calculated Cohen’s d effect size. The text is updated accordingly. P is provided for the standardized beta by using *p<0.05, **p< 0.01, ***p<0.001. I hope this makes the tables more comprehensive.

  1. The names of tables 2, 3 and 4 do not correspond to what I see in them.

Response: The names have been updated accordingly to correspond better with what you can see in them:

 “Table 2. Linear regression analyses for the organizational factors on the different burnout measures (BAT and COVID-19-BS)”

“Table 3. Linear regression analyses for the situational (COVID-19) factors on the different burnout measures (BAT and COVID-19-BS)”

“Table 4. Linear regression analyses for the psychological factors on the different burnout measures (BAT and COVID-19-BS)”

  1. You need to have less hypothesis. In fact only one!

Response: As I am testing several hypotheses, I find it hard to reduce them more. I have tried to narrow down the hypotheses by using (a), (b), (c). In example “The COVID-19-specific demands of (a) involvement with COVID 19, (b) fear of COVID 19, and (c) COVID 19-related stress are positively correlated with burnout.”

Round 2

Reviewer 1 Report

Thanks for the revision.  I only have one further comment for question 4 in the first round of review. 

I understand you categorized some variables and coded them by levels.

For example, the employment status was coded as 1 (1–50%), 2 (51–99%), or 3 (100%) was coded by 3 levels.  When analyzing the categorical variable, contrast is usually used. That is, we are comparing levels to the reference level.  Then we will have 3-1=2 betas estimated.  But in your table, there is only one beta for this variable.  To me, it means you treated 'employment status' as a continuous variable with values 1 to 3, instead of levels, when you fit your model by using the software.   Let's put it in another way, how do you interpret the \beta in your table for this variable? I understand it is just an adjustment covariate.  Thanks. 

Author Response

Dear reviewer

Thank you for your clarification of this request. As I understand you ask me to dummy code the background variables and use a reference category. Employment status and age are ordinal variables and treated as continuous variables in the analyses. To clarify this, I have changed the text accordingly and corrected categorial to ordinal. This implies that a positive beta, means an increase in these variables. I hope this clarifies.

Alternatively, I can omit employment status and redo all the analyses.

Reviewer 2 Report

Dear Authors,

Congratulations on the work done to correct and improve your manuscript. Now, I do consider that it can be accepted for publication.

Kind regards.

Author Response

thank you:-)

Reviewer 3 Report

Thanks author for accepting required correction. 

Author Response

Thank you